# Flow-Attention-based Spatio-Temporal Aggregation Network for 3D Mask Detection

**Yuxin Cao**[*]
Tsinghua University
China

**Yian Li**[*]
ShanghaiTech University
China

**Yumeng Zhu**
Ping An Technology
(Shenzhen) Co., Ltd.
China

**Derui Wang**
CSIRO's Data61
Australia

**Minhui Xue**
CSIRO's Data61
Australia

## Abstract

Anti-spoofing detection has become a necessity for face recognition systems due to the security threat posed by spoofing attacks. Despite great success in traditional attacks, most deep-learning-based methods perform poorly in 3D masks, which can highly simulate real faces in appearance and structure, suffering generalizability insufficiency while focusing only on the spatial domain with single frame input. This has been mitigated by the recent introduction of a biomedical technology called rPPG (remote photoplethysmography). However, rPPG-based methods are sensitive to noisy interference and require at least one second (> 25 frames) of observation time, which induces high computational overhead. To address these challenges, we propose a novel 3D mask detection framework, called FASTEN (Flow-Attention-based Spatio-Temporal aggrEgation Network). We tailor the network for focusing more on fine-grained details in large movements, which can eliminate redundant spatio-temporal feature interference and quickly capture splicing traces of 3D masks in fewer frames. Our proposed network contains three key modules: 1) a facial optical flow network to obtain non-RGB inter-frame flow information; 2) flow attention to assign different significance to each frame; 3) spatio-temporal aggregation to aggregate high-level spatial features and temporal transition features. Through extensive experiments, FASTEN only requires five frames of input and outperforms eight competitors for both intra-dataset and cross-dataset evaluations in terms of multiple detection metrics. Moreover, FASTEN has been deployed in real-world mobile devices for practical 3D mask detection.

## 1 Introduction

Face recognition has been deeply involved in various prevalent applications, such as access control systems and e-commerce [1]. Nevertheless, similar to the potential security threat in fields of images [2, 3] and streaming videos [4, 5], face recognition systems are encountered with security issues mainly caused by face presentation attacks in the forms of printed photos, replayed videos, 3D masks [6, 7]. Since face images are easily accessible on social networks, one can mass-produce spoofing faces at low acquisition and manufacturing costs. To mitigate the detriment caused by face presentation attacks, a plethora of face presentation attack detection (PAD) methods have sprung up in the last decade [8–18]. Despite the acceptable detection performance against printings/screens, 3D masks can undermine these PAD methods by a large margin due to advanced 3D printing

---

[*]This work was done while Yuxin and Yian worked as interns at Ping An Technology (Shenzhen) Co., Ltd.

37th Conference on Neural Information Processing Systems (NeurIPS 2023).

technologies [19, 20]. 3D masks made up of soft silicone are vivid enough, in terms of color, texture, and geometry structure, that even humans find it confusing to differentiate. This has gradually made 3D mask defense from a branch of face anti-spoofing to a relatively independent sub-topic [19, 21, 22].

Existing 3D mask PAD methods can be categorized into three branches: Handcrafted-Features-based methods (HFMs) [9–12, 23], Deep-Learning-based methods (DLMs) [15–18] and remote-Photoplethysmography-based methods (rPPGMs) [21, 24–28]. In earlier years, 3D masks were mainly distinguished by manually annotated features. In the last decade, many works have greatly improved the performance of spoofing detection through the powerful advantages of deep learning. However, HFMs and DLMs do not fully consider temporal differences, resulting in average effectiveness in cross-dataset evaluation. There has been a new breakthrough with the emergence of rPPG technology, which captures the color variations caused by heartbeat when the face skin is illuminated by light. rPPGMs achieve splendid performance on cross-dataset evaluation, but rPPG signals are touchy about noise interference [28]. The release of the HiFiMask dataset [29] also challenges rPPGMs since the light frequency they used is similar to that of the heartbeat, leading to loss of utility of rPPG signals. We summarize the limitations of these methods as follows. 1) Data availability. Most methods obtain data from devices that are difficult to deploy in various practical scenarios, *e.g.,* depth information captured by RGBD cameras [30], rPPG signals obtained by light [27]. 2) Temporal information involvement. Few works focus on temporal features, which can carry variations of facial details. 3) Computational cost. The shortest observation time required for rPPGMs is one second (> 25 frames), which is still long enough to trigger a high computational cost.

To address the limitations above, we propose FASTEN (Flow Attention-based Spatio-Temporal aggrEgation Network), which only needs five frames of the input RGB video (no depth, rPPG, or other information is needed) to distinguish between 3D masks and real faces. FASTEN fully utilizes temporal and spatial information by focusing more on fine-grained details in large movements, which can avoid redundant feature interference and lower the computational cost. We consider frame-wise attention to aggregate the temporal and spatial feature information of multiple frames, rather than simple concatenation, since the latter practice neglects the frame significance discrepancy, especially when the input frames include large movements, *e.g.,* eye blinking and mouth opening. Concretely, we first train a facial optical flow network, named FlowNetFace, to obtain RGB-free inter-frame flow information. The frame weights are then calculated by facial optical flow and assigned to each frame as their attention scores. Finally, the high-level spatial features and temporal transition features are aggregated for detection. In summary, the contributions of this paper are listed as follows.

- We propose a flow-attention-based spatio-temporal aggregation network, called FASTEN, for 3D mask PAD. FASTEN outperforms existing works in intra-/cross-dataset evaluations.
- In order to thoroughly take advantage of the temporal information of spatial features, we train a facial optical flow model, FlowNetFace, to learn the small variations in facial features between frames. The facial features are applied simultaneously to the calculation of frame weights and temporal transitional features.
- We address the long-time issue that the state-of-the-art rPPGM still needs about one second (> 25 frames) for observation, and propose a frame-based method that only needs any five frames of a 3D mask video, by adding frame weights to multi-frame spatial features. We also release a lightweight application for FASTEN to mobile devices, showing that FASTEN can achieve practical 3D mask PAD on resource-constrained platforms.

## 2   Related work

**Handcrafted-Features-based Methods.** In early studies, due to the conspicuous artifacts in presentation attacks such as printings and screens, facial characteristics and handcrafted descriptors, such as LBP [31, 32], HOG [8] and SIFT [33], were utilized to distinguish spoofing images. These characteristics, however, are not adaptable to 3D masks, since masks do not incorporate face quality defects such as Moiré patterns [34] or reflection [35]. Analyzing color texture [9] in chroma images is also a common kind of feature to detect color reproduction artifacts in masks. From the perspective of dynamic features, facial cues such as eye-blinking [10, 23] and face movement [11, 12] are commonly used to discriminate printings and screens, but are not congenial to 3D masks [36].

**Deep-Learning-based Methods.** In recent years, a plethora of DLMs [15–18, 37–39] have burst out. CDCN++ [15] uses Central Difference Convolution (CDC) to learn central gradient information after

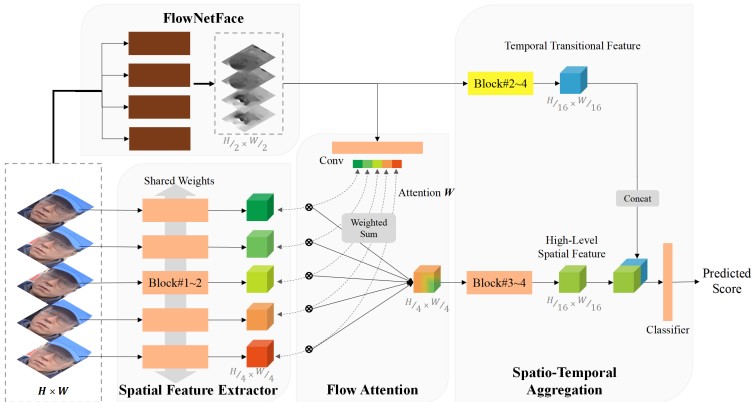

Figure 1: Overview of our proposed FASTEN.

which a multi-scale attention module is added to fuse multi-scale features. Due to its practicality and popularity, some follow-up work [40, 41] is based on CDCN++. HRFP [37] separates faces into several parts to extract fine-grained information. It came out first in a 3D high-fidelity mask attack detection challenge held by the International Conference on Computer Vision (ICCV) Workshop in 2021. Temporal Transformer Network (TTN) [30] has recently been proposed by learning subtle temporal differences from depth maps captured by RGBD cameras. The depth maps of all kinds of spoofing images are set as 0 during training. However, 3D masks actually contain depth information and cannot be ignored, leading to mediocre detection ability towards 3D masks. Besides, depth information is hard to obtain in practical scenarios. Thanks to the weak generalizability of deep learning networks, DLMs find it tough to detect unseen 3D masks.

**rPPG-based Methods.** Emerging in recent years, rPPG achieves non-contact monitoring of human heart activity by using RGB cameras to detect subtle color variations in facial skin caused by heartbeat under room light [42, 43]. Since light transmission is blocked in 3D masks, the heartbeat can only be detected from real faces [44]. As a result, rPPGMs achieve much success when distinguishing 3D masks [21, 24–28]. Among advanced work, MCCFrPPG [28] improves CFrPPG [44] by extracting temporal variation from rPPG signals through a multi-channel time-frequency analysis scheme to mitigate interference of noise on rPPG signals. To shorten the required observation time from 10 seconds to within one second, LeTSrPPG [21] evaluates the temporal similarity of the rPPG features and proposes a self-supervised-learning-based spatio-temporal convolution network to distinguish 3D masks in terms of the consistency of the rPPG signals. However, a one-second video still includes many frames (> 25) and requires a significant amount of computation. The HiFiMask dataset [29] also challenges rPPGMs since the light frequency they used is similar to that of heartbeat, leading to loss of distinguishment utility of rPPG signals.

## 3 Methods

Given a video $v = [x_1, x_2, ..., x_T]$ containing $T\ (T \geq \lambda)$ frames of a real face or a face wearing a 3D mask, our goal is to build a discriminative network that can accurately distinguish 3D masks from real faces, where $\lambda$ denotes the required input frame number (5 in our case). Different from the existing 3D mask detection method, FASTEN relaxes the restrictions on the input, *i.e.,* we only need any $\lambda$ frames in sequential order from the video $x = [x_{a_1}, x_{a_2}, ..., x_{a_\lambda}], 1 \leq a_1 < a_2 < ... < a_\lambda \leq T$, where the selected adjacent frames can be either equally spaced or unequally spaced. The framework of FASTEN is depicted in Figure 1. In our design, three key components are integrated into the backbone network: 1) We first employ an optical flow model FlowNetFace to generate facial optical flow between adjacent frames. 2) The flow attention module is used to calculate the frame weights over time and assign different attention to each frame. 3) Finally, the temporal transition features are extracted from FlowNetFace and aggregated with the high-level spatial features for detection.

### 3.1 Feature extractor

**Spatial features** capture facial details, especially information on facial texture, which is conducive to 3D mask detection. We use $0.75\times$ MobileNetV3-Small [45], which contains four blocks, as our

backbone to balance accuracy and efficiency. Considering that it is rather computationally expensive if low-level spatial features $F = [F_{a_1}, F_{a_2}, ..., F_{a_\lambda}]$ are in high resolution, the low-level spatial features of $\lambda$ frames are extracted from the first two stages of MobileNet with shared network weights. Also, the receptive field is 19 after the first two blocks. Corresponding to our input face region with a resolution of $96 \times 112$, it is just right at this stage to extract the features of a specific part (*e.g.,* eyes or corners of mouth), hence the fusion of multi-frame spatial features can be better aligned.

**Temporal features** encode the motion patterns of facial muscles, which is beneficial to distinguish fake faces bearing a strong resemblance to real ones in appearance. We choose facial optical flow maps to represent temporal features based on the following insights: 1) Unlike rPPG, optical flow, which embeds the motion speed and direction of each pixel, can preserve the facial structure while partially neglecting the interference caused by color diversity. This helps to withstand multiple challenges of robustness such as acquisition equipment and facial skin color. 2) By comparing the optical flow of every two adjacent frames in a clip, we can identify moments with larger movement and more exposed traces, guiding the network to allocate attention reasonably and thereby reducing the demand for frame numbers.

### 3.2 Facial optical flow network

Since there are no existing optical flow models specifically designed for faces, we propose an active-learning-based alternative strategy to train a facial optical flow prediction network ourselves. As one of the classic methods in the field of optical flow, FlowNet [46] generally contains an encoder/decoder structure and works well on the proposed generic optical flow dataset named Flying Chairs. The variants of FlowNet include FlowNetS, FlowNetC, FlowNet2.0, *etc*, whose performance improves gradually as well as computational complexity. Unlike the Flying Chairs dataset, which is composed of high-resolution images with multiple targets, face images are rather simple. Therefore, we build a facial optical flow model evolved from FlowNetS by reducing the layer number to 2 in both the encoder and the decoder. In this case, our optical flow model, namely FlowNetFace, can obtain the temporal features required at the lowest computational cost. In addition, we use FlowNet2.0 [47], a stacked deep-learning-based optical flow estimator, to generate the ground truth of the facial optical flow. Following prior work [46, 47], the average endpoint error (AEPE) is used to measure the distance between the predicted and ground truth optical flow vectors.

### 3.3 Flow attention

The most significant difference between 3D masks and real faces is basically the splicing traces on the face. Especially, when a person is asked to blink the eyes or open the mouth, the unnatural details are particularly noticeable, *e.g.,* compared to static faces, the detailed features around the eyes when eyes are closed are more helpful in distinguishing masks from real faces. As aforementioned, mask-splicing clues change with facial expressions. Therefore, scoring spatial features through temporal information can make the network autonomously attend to significant frames, thereby amplifying the features of facial details under special actions as well as weakening interfering information. Concretely, we input optical flow of the selected $\lambda$ frames, sequenced by a $3 \times 3$ convolution layer to calculate the frame weights $W = [w_{a_1}, w_{a_2}, ..., w_{a_\lambda}]$ for the spatial features of each frame. Then, the frame weights are fused to the spatial features $F$, and the output fused feature can be expressed as

$$F_{fuse} = \sum_{i=1}^{\lambda} w_{a_i} F_{a_i}, \quad \text{s.t.} \sum_{i=1}^{\lambda} w_{a_i} = 1. \tag{1}$$

### 3.4 Spatio-temporal aggregation

As for spatial information, the fused feature $F_{fuse}$ is then fed into the latter two blocks of the backbone to obtain the spatial feature $F_{spa}$. It is apparent that there exist smooth facial fluctuations and muscle movements in real faces when a video is captured, while no such phenomena are presented in 3D masks. Thus, we additionally consider extracting the temporal transitional feature $F_{temp}$ that can represent the temporal differences between consecutive frames. Since the resolution of the optical flow features is eight times that of $F_{spa}$, we input them into the module composed of the latter three blocks of MobileNet, resulting in temporal features of the same size as deep-level spatial features. Since the spatial feature can reflect the splicing traces in the input face image, and the temporal transitional feature brings the transitional movement cues, we aggregate both the spatial and temporal features by concatenation and then input the aggregated into the classifier which contains a fully

connected layer followed by a Softmax layer. Due to the data imbalance between real faces and 3D masks, we use the weighted BCE loss to train the network. Given a batch of $N$ face images, we have

$$\mathcal{L} = \frac{1}{N} \sum_{i=1}^{N} [-\mu y_i \log p_i - (1 - y_i) \log (1 - p_i)], \tag{2}$$

where $y_i$ denotes the true class of the $i$-th image, with 1 standing for real faces, and 0 representing 3D masks. $p_i$ denotes the output score of the classifier. $\mu$ denotes the balancing weight.

## 3.5  Training strategy

Considering that the proposed FASTEN network includes several modules, the training strategy is given as follows. 1) We first separately train the facial optical flow network FlowNetFace using the face dataset and the ground truth flow obtained by FlowNet2.0 [2]. Since optical flow prediction is independent of detection, training FlowNetFace alone can not only converge the network as soon as possible but also avoid conflicts caused by two different tasks. 2) Then we use the MobileNetV3-Small model pre-trained on ImageNet [3] as our backbone. We finally use a small learning rate to finetune the entire network on 3D mask datasets, with a smaller learning rate for FlowNetFace. Please refer to our code for more details [4].

# 4  Experiments

## 4.1  Experimental setup

**Datasets.** We conduct our experiments on three frequently used datasets, 3DMAD (3D Mask Attack Dataset) [19], HKBU-MARs V1+ (Hong Kong Baptist University 3D Mask Attack with Real-World Variations) [48] and HiFiMask (High-Fidelity Mask dataset) [29]. 1) 3DMAD [19] contains 255 videos of 17 subjects wearing custom-built Thatsmyface masks where the eyes and nose holes are cut out to present more natural wearing effects. The videos are recorded by a Kinetic camera at $640 \times 480$ resolution, 30 fps. Both color and depth data are collected. 2) HKBU-MARs V1+ [48] includes 180 videos of 12 subjects with two mask types, Thatsmyface and REAL-f. Videos in HKBU-MARs V1+ are captured by Logitech C920 under room light. 3) HiFiMask [29] is a large, high-resolution mask dataset containing 54,600 videos from 75 subjects wearing three types of masks: plaster masks, resin masks, and transparent masks. HiFiMask considers six different scenes, including light variation and head motion. The frequency of the periodic flashlight is set within $[0.7, 4]$ Hz, which is consistent with that of ordinary people's heartbeat, to a great extent challenging rPPGMs since pseudo 'liveness' signals can be obtained for masks. More details of the datasets are shown in Table 1. HiFiMask is the most challenging dataset for PAD methods, owing to its large quantity and high fidelity.

**Competitors.** We select eight existing detection methods for spoofing images or specifically for 3D masks from HFMs, DLMs, and rPPGMs as our competitors, *i.e.*, MS-LBP [32], CTA [9], CDCN++ [15], HRFP [37], ViTranZFAS [38], MD-FAS [39], CFrPPG [44], and LeTSrPPG [21]. In the early ages, MS-LBP [32] and CTA [9], as two classic feature descriptors, exhibit edges in finding quality defects in terms of color and texture. Although most of the DLMs are not tailored for 3D masks, we select CDCN++ [15], one of the most popular and frequently used deep-learning networks, to represent this branch of detection methods, since the fine-grained features of spoofing faces learned by central difference convolution and multi-scale attention module are expected to work for 3D masks. HRFP [37] was crowned with laurel in the 3D high-fidelity mask attack detection challenge held by ICCV in 2021. By segmenting faces into several parts for feature extraction, HRFP rejects the input face once an anomaly exists in any part of the input. Two rPPGMs, CFrPPG [44] and LeTSrPPG [21], are also included for comparison due to their strong performance in identifying 3D masks based on biotechnology. We further consider two DLMs using pre-trained models due to their outstanding performance in face anti-spoofing tasks. ViTranZFAS [38] used transfer learning from a pre-trained vision transformer (ViT) to distinguish spoofing images. MD-FAS [39] was later proposed to improve domain adaptation by only using target domain data to update the pre-trained face anti-spoofing model. For all the competitors, we follow their original settings.

---

[2] https://github.com/NVIDIA/flownet2-pytorch
[3] https://github.com/d-li14/mobilenetv3.pytorch
[4] https://github.com/JosephCao0327/FASTEN

Table 1: Concrete details of different datasets we used. The resolution of HiFiMask varies.

| Dataset | Year | #Subject | #Mask | Mask type | Scenes | Light | Camera | #Live videos | #Masked videos | #Total videos | Resolution | Facial resolution |
|---|---|---|---|---|---|---|---|---|---|---|---|---|
| 3DMAD [19] | 2013 | 17 | 17 | 2 | 1 | 1 | 1 | 170 | 85 | 255 | $640 \times 480$ | $80 \times 80$ |
| HKBU-MARs V1+ [48] | 2018 | 12 | 12 | 2 | 1 | 1 | 1 | 120 | 60 | 180 | $1280 \times 720$ | $200 \times 200$ |
| HiFiMask [29] | 2021 | 75 | 75 | 3 | 6 | 6 | 7 | 13,650 | 40,950 | 54,600 | High fidelity | $400 \times 400$ |

**Statement for rPPGMs.** Due to the lack of open-source codes for rPPGMs, we are unable to obtain rPPG signals to replicate their method. Thus, we cite the results reported in their paper on 3DMAD and HKBU-MARs V1+. We do not involve one recent work MCCFrPPG [28] since the observation time is too long (10∼12 seconds) to be fairly compared with our method, which only needs five frames. Instead, we select its previous version CFrPPG [44] since the detection results with an observation time of one second are available in the paper of LeTSrPPG [21]. Note that HiFiMask only contains 10 frames sampled from each video at equal intervals, the sampling frequency of which is much lower than twice the highest light frequency (4Hz for periodic flashlight). According to the Nyquist-Shannon Sampling Theorem [49], recovering rPPG analog signals will inevitably lead to frequency aliasing and signal distortion. Ergo, we do not consider rPPGMs on HiFiMask.

**Metrics.** Following prior work [15, 21, 28, 37], we use five metrics to evaluate the detection performance, *i.e.,* Half Total Error Rate (HTER), Equal Error Rate (ERR), Area Under Curve (AUC), Attack Presentation Classification Error Rate (APCER), and Bonafide Presentation Classification Error Rate (BPCER). We use "B@A=0.1/0.01" to represent the BPCER when APCER = 0.1/0.01.

**Parameter settings.** In the training process, we adopt commonly utilized augmentations including Gaussian noise, random brightness contrast, and cut-out. FlowNetFace is trained (finetuned) using AdamW optimizer with a learning rate of 1e-4 (2e-5) and weight decay as 4e-4 (1e-2 for 3DMAD and HKBU-MARs V1+, and 1e-3 for HiFiMask). The rest of the network is finetuned using the same optimizer with a learning rate of 2e-4, which is adjusted by cosine annealing with warm restarts. We set the warm-up epoch number as 2 and the total epoch number as 150. We set the required input frame number $\lambda = 5$, the balancing weight $\mu = 0.5$ for 3DMAD and HKBU-MARs V1+ and $\mu = 3$ for HiFiMask. The batch size is 120, and all models are trained using two Tesla V100 GPUs. We perform face alignment [50] on images before inputting them into the network, so that spatial features can be fused without misalignment.

### 4.2 Experimental results

**Intra-dataset evaluation.** For the intra-dataset evaluation, we adopt leave one out cross-validation (LOOCV) protocol on 3DMAD and HKBU-MARs V1+. Following [28], a test subject is randomly selected first for both two datasets. Then we randomly select eight/five subjects for training and the leftover eight/six subjects for validation on 3DMAD/HKBU-MARs V1+. LOOCV is not suitable for large datasets such as HiFiMask, since LOOCV is subject to high variance or overfitting [51]. Instead, we use *k*-fold cross validation (KCV) when training on HiFiMask. In this case, we follow Protocol 1 [29], where 45, 6 and 24 subjects are selected for training, validation and testing, respectively.

The intra-dataset evaluation results reported in Table 2 show that FASTEN performs better than the other eight competitors, with AUCs of 99.8%, 99.7%, and 99.8%, and HTERs of 1.17%, 2.13% and 2.11% on 3DMAD, HKBU-MARs V1+ and HiFiMask, respectively. We attribute the outperformance to FASTEN's strong ability to distinguish 3D masks by assigning different weights to each frame and aggregating both spatial and temporal information. MS-LBP ranks second on 3DMAD since handcrafted features perform well under fixed lighting conditions. However, the performance of MS-LBP is limited in complex lighting and different mask types. HRFP comes out second and third in both HKBU-MARs V1+ and HiFiMask since the fine-grained features of face parts at high resolution in HRFP help identify subtle differences in faces. CDCN++ also performs well owing to the multi-scale attention which can effectively aggregate multi-level features of central difference convolution. We do not see significant advantages for rPPGMs due to noise interference by various lighting conditions and camera settings. As previously discussed, rPPGMs are challenged by HiFiMask, which may induce a loss of utility.

**Cross-dataset evaluation.** We also carry out a cross-dataset evaluation to verify the generalizability of FASTEN when test scenarios have never been seen before. Models are trained on one dataset and then tested on the test set of another dataset. Table 3 reports the cross-dataset evaluation results. We first consider the scenarios between two small datasets, 3DMAD and HKBU-MARs V1+. The performance of HFMs drops dramatically due to the limitations of simple color and texture features.

Table 2: Intra-dataset evaluation on different datasets. 'U': unavailable. 'N/A': not applicable.

| Method | 3DMAD | | | | | HKBU-MARs V1+ | | | | | HiFiMask (Protocol 1) | | | | |
|---|---|---|---|---|---|---|---|---|---|---|---|---|---|---|---|
| | HTER (%) | EER (%) | AUC | B@A =0.1 | B@A =0.01 | HTER (%) | EER (%) | AUC | B@A =0.1 | B@A =0.01 | HTER (%) | EER (%) | AUC | B@A =0.1 | B@A =0.01 |
| MS-LBP [32] | 1.92±3.4 | 3.28 | 99.4 | 0.32 | 6.78 | 24.0±25.6 | 22.5 | 85.8 | 48.6 | 95.1 | 48.3 | 40.5 | 63.7 | 78.9 | 97.3 |
| CTA [9] | 4.40±9.7 | 4.24 | 99.3 | 1.60 | 11.8 | 23.4±20.5 | 23.0 | 82.3 | 53.8 | 89.2 | 40.7 | 31.6 | 74.9 | 64.1 | 92.9 |
| CDCN++ [15] | 4.20±7.1 | 8.34 | 96.7 | 6.62 | 59.6 | 4.83±7.6 | 8.70 | 96.0 | 7.77 | 66.2 | 3.67 | 2.64 | 99.6 | 0.83 | 6.79 |
| HRFP [37] | 2.34±5.2 | 2.41 | 99.7 | 0.67 | 5.87 | 3.34±5.7 | 4.33 | 99.2 | 1.34 | 6.23 | 2.20 | 2.26 | 99.7 | 0.28 | 4.35 |
| ViTranZFAS [38] | N/A | N/A | N/A | N/A | N/A | N/A | N/A | N/A | N/A | N/A | 2.63 | 2.48 | 99.7 | 0.21 | 6.58 |
| MD-FAS [39] | N/A | N/A | N/A | N/A | N/A | N/A | N/A | N/A | N/A | N/A | 5.45 | 4.12 | 99.4 | 0.34 | 15.2 |
| CFrPPG (1s) [28] | 32.7±7.4 | 32.5 | 70.8 | U | U | 42.1±5.6 | 42.0 | 60.8 | U | U | N/A | N/A | N/A | N/A | N/A |
| LeTSrPPG [21] | 11.8±8.6 | 11.9 | 94.4 | U | U | 15.8±6.5 | 15.7 | 91.5 | U | U | N/A | N/A | N/A | N/A | N/A |
| FASTEN(Ours) | 1.17±0.7 | 1.06 | 99.8 | 0.03 | 0.31 | 2.13±5.1 | 2.56 | 99.7 | 0.81 | 2.83 | 2.11 | 2.18 | 99.8 | 0.13 | 4.13 |

Table 3: Cross-dataset evaluation. 'U': unavailable. 'N/A': not applicable.

| Method | 3DMAD → HKBU-MARs V1+ | | | | | HKBU-MARs V1+ → 3DMAD | | | | | HiFiMask → 3DMAD | | | | | HiFiMask → HKBU-MARs V1+ | | | | |
|---|---|---|---|---|---|---|---|---|---|---|---|---|---|---|---|---|---|---|---|---|
| | HTER (%) | EER (%) | AUC | B@A =0.1 | B@A =0.01 | HTER (%) | EER (%) | AUC | B@A =0.1 | B@A =0.01 | HTER (%) | EER (%) | AUC | B@A =0.1 | B@A =0.01 | HTER (%) | EER (%) | AUC | B@A =0.1 | B@A =0.01 |
| MS-LBP | 47.7±7.0 | 48.3 | 52.4 | 86.4 | 97.6 | 43.2±7.3 | 43.7 | 58.8 | 87.5 | 99.2 | 47.8 | 41.3 | 62.6 | 76.3 | 98.0 | 50.4 | 57.0 | 39.1 | 93.0 | 98.8 |
| CTA | 51.5±2.4 | 55.3 | 48.9 | 90.5 | 98.8 | 68.2±7.7 | 65.4 | 40.1 | 94.7 | 99.9 | 50.1 | 71.5 | 32.2 | 97.6 | 99.8 | 49.5 | 63.3 | 42.9 | 90.0 | 94.5 |
| CDCN++ | 50.3±2.7 | 55.3 | 42.3 | 93.2 | 99.9 | 41.1±6.8 | 33.1 | 66.2 | 74.1 | 98.9 | 36.8 | 37.7 | 72.1 | 65.9 | 87.1 | 35.7 | 29.7 | 80.8 | 52.9 | 78.4 |
| HRFP | 25.8±3.1 | 32.3 | 69.2 | 81.3 | 98.4 | 6.76±0.9 | 7.25 | 98.6 | 5.63 | 22.0 | 21.3 | 12.4 | 95.8 | 14.6 | 32.5 | 7.68 | 8.33 | 99.1 | 1.52 | 11.4 |
| ViTranZFAS | N/A | N/A | N/A | N/A | N/A | N/A | N/A | N/A | N/A | N/A | 26.3 | 16.5 | 91.1 | 23.1 | 60.8 | 32.7 | 20.6 | 88.3 | 34.2 | 60.2 |
| MD-FAS | N/A | N/A | N/A | N/A | N/A | N/A | N/A | N/A | N/A | N/A | 33.5 | 34.4 | 69.9 | 67.0 | 75.3 | 9.38 | 8.61 | 97.4 | 5.67 | 43.4 |
| CFrPPG (1s) | 39.2±1.4 | 40.1 | 63.6 | 75.5 | U | 40.1±2.3 | 40.6 | 62.3 | 79.1 | U | N/A | N/A | N/A | N/A | N/A | N/A | N/A | N/A | N/A | N/A |
| LeTSrPPG | 15.7±0.5 | 16.6 | 90.1 | 25.2 | U | 12.9±0.8 | 13.1 | 93.4 | 15.8 | U | N/A | N/A | N/A | N/A | N/A | N/A | N/A | N/A | N/A | N/A |
| FASTEN(Ours) | 11.8±2.1 | 16.8 | 91.0 | 18.8 | 24.4 | 3.85±1.9 | 4.35 | 99.1 | 3.00 | 8.88 | 2.35 | 2.71 | 99.8 | 0.27 | 4.76 | 7.38 | 8.55 | 98.3 | 5.40 | 17.7 |

Despite the strong performance in intra-dataset, DLMs exhibit a significant decline in performance, *e.g.,* the AUC for CDCN++ on 3DMAD drops from 96.7 to 42.3 when testing HKBU-MARs V1+. Such results are attributed to the fact that DLMs lack generalization when encountered with unseen inputs. Another major reason why FASTEN outperforms these methods in cross-dataset evaluation could be the aggregation between spatial and temporal features, the latter of which carries a large amount of interframe information and can better capture the flaws in splicing traces. Although rPPGMs have been verified to have strong generalization ability, their accuracy is easily affected by noise and light frequency, resulting in slightly lower performance than FASTEN.

Due to the large gap between the two small datasets and HiFiMask in data quantity, we discard the evaluation from small datasets to HiFiMask. Conversely, we conduct cross-dataset evaluation when the model is trained on HiFiMask, and test on 3DMAD and HKBU-MARs V1+. FASTEN achieves over 98% AUCs with HTER below 8% in both scenarios. As for rPPGMs, they cannot be implemented on HiFiMask due to the sampling problem. Since the light frequency is similar to that of heartbeat, the rPPG signals of 3D masks contain pseudo 'liveness' cues in both time and frequency domains, misguiding rPPGMs to incorrectly classify 3D masks as real faces. Therefore, we point out that their performance could be very limited, even if the proposer of HiFiMask [29] provided complete videos before sampling and the recovered rPPG signals. Overall, the outperformance of FASTEN shows the generalizability of our proposed framework on both small and large datasets.

**Discussion of ViTranZFAS and MD-FAS.** ViTranZFAS requires large amount of data for training, while MD-FAS focuses more on domain adaptation, whose experimental setting is slightly different from 3D mask defense. Therefore, we only conduct experiments on HiFiMask. We attribute the poor performance of ViTranZFAS to three main reasons: 1) Lack of temporal information: ViT focuses primarily on analyzing individual images without considering temporal cues, but 3D mask detection often requires understanding the dynamic changes over time. 2) Small regions of interest: Liveness cues often involve subtle facial regions, such as the edge of facial features or muscle contractions. However, ViT's patch-based approach may not focus on these small regions of interest effectively, leading to challenges in capturing fine-grained liveness indicators. 3) Training difficulty: ViT requires a pre-trained model and high computational costs. MD-FAS aims to solve the forgetting issue when learning new domain data and improve the adaptability of the model. It cannot work well with unseen target data, as shown in Table 3. In addition, the baseline SRENet selected by this scheme is a region-based fine classification task, which has high requirements for the diversity of attack categories, as well as the resolution of images. When we use HiFiMask as a training set for a fair comparison, its accuracy performance remains poor.

**Overall evaluation.** We calculate the average AUC, ERR, and HTER of all the PAD methods in both intra-dataset and cross-dataset evaluations to compare the overall performance. The results in

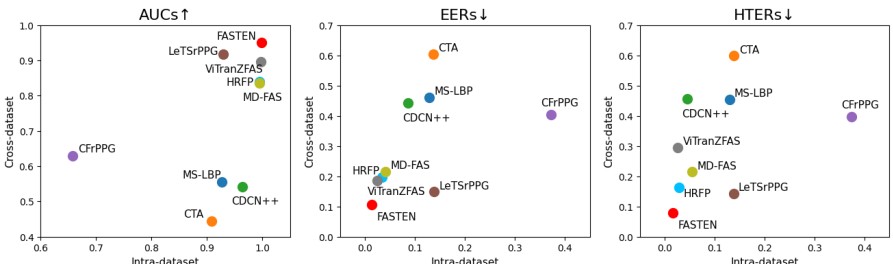

Figure 2: Comparison of both intra and cross-dataset evaluations.

Figure 2 demonstrate that simultaneously considering temporal and spatial features helps FASTEN achieve the best detection performance and generalizability.

**Computational overhead.** We further compare the computational cost of different PAD methods when detecting 3D masks. Since MS-LBP and CTA only need handcrafted features for detection and rPPGMs require long videos with rPPG signal extraction, we only compare CDCN++ and HRFP with our proposed method under the same inputs. The average inference time of CDCN++, HRFP and FASTEN is 28.8ms, 286.5ms and 25.3ms respectively. Compared to HRFP, FASTEN reduces computational cost and saves inference time with the benefit of smaller networks. Although CDCN++ achieves a comparable time cost with FASTEN, the lack of consideration of temporal features leads to insufficient detection performance. Compared with the best rPPGMs, LeTSrPPG which requires one second (> 25 frames) for observation, FASTEN only needs five frames as input, which can significantly reduce computational overhead.

## 4.3 Ablation study

**Effectiveness of components.** There are two main components in FASTEN, the spatial branch and the temporal branch. To verify the effectiveness of each component, we consider the following five scenarios. 1) Only spatial features considered (Spat.): the temporal branch, including optical flow prediction, frame weight assignment, and temporal transitional feature extraction, is removed. 2) Only temporal features considered (Temp.): the spatial feature extraction and frame weight assignment are removed. 3) Equal frame weights (Spat. + Temp.(Eq.)): the frame weight assignment is replaced by equally assigning frame weights ($\forall i \in \{1, 2, ..., \lambda\}, w_{a_i} = \frac{1}{\lambda}$) to each frame. 4) Different frame weights (Spat. + Temp.(UnEq.), also FASTEN): FASTEN under its default settings. 5) The FlowNetFace is frozen when finetuning the network (Frozen FlowNetFace). Table 4 reports the intra-dataset evaluation on HiFiMask. Similar to most existing DLMs, only considering spatial information neglects the subtle transitional features and unnatural prosthetic traces, resulting in higher HTER (3.11%). When only temporal information is considered, the trained model takes no notice of the splicing traces and unnatural gloss on the masks when the subject is closing their eyes or opening their mouth. Therefore, detection performance drops dramatically (with 7.47% HTER). When all frames are weighted equally, assigning lower weights to frames with obvious actions (*e.g.,* eye-blinking) also leads to performance abatement. This also indicates that FASTEN performs better than simply concatenating temporal and spatial features.

The performance gap between the first three scenarios and FASTEN demonstrates that fully and reasonably utilizing spatial and temporal features is beneficial for improving detection accuracy. Intuitively, the network might perform better if FlowNetFace is frozen when the network is finetuned, as the optical flow seems to remain fixed when the network is learning the distinguishment between 3D masks and real faces. The experimental results in the last scenario nevertheless show that slightly adjusting FlowNetFace with a minuscule learning rate (2e-5) can improve the detection performance. We draw a conclusion that the joint finetuning of the optical flow model and detection model is instrumental in improving the supervision of temporal features over spatial features, thereby providing more adaptive spatial information for detection.

**Possible variants of FASTEN.** Instead of fixing the input frame number $\lambda$ as 5, the number could be varied. We conduct an ablation study on HiFiMask under different input frame numbers, and the results are also reported in Table 4. Since the total frame number of HiFiMask is 10, we set $\lambda$ as 2, 5, 8 and 10. We follow the same frame selection strategy as FASTEN, where the $\lambda$ frames are randomly selected and sorted in sequential order. When $\lambda = 2$, the lack of optical flow information results in

Table 4: Ablation study of FASTEN on HiFiMask.

| Component | HTER (%) | EER (%) | AUC | B@A =0.1 | B@A =0.01 | $\lambda$ | HTER (%) | EER (%) | AUC | B@A =0.1 | B@A =0.01 |
|---|---|---|---|---|---|---|---|---|---|---|---|
| Spat. | 3.11 | 3.06 | 99.64 | 0.58 | 7.28 | 2 | 3.73 | 3.93 | 99.55 | 0.62 | 8.48 |
| Temp. | 7.47 | 7.51 | 98.15 | 5.56 | 24.8 | 5 | 2.11 | 2.18 | 99.85 | 0.13 | 4.13 |
| Spat. + Temp.(Eq.) | 2.91 | 2.89 | 99.67 | 0.52 | 6.76 | 8 | 2.78 | 2.76 | 99.73 | 0.52 | 7.34 |
| Spat. + Temp.(UnEq.) | 2.11 | 2.18 | 99.85 | 0.13 | 4.13 | 10 | 2.90 | 3.01 | 99.55 | 0.71 | 7.25 |
| Frozen FlowNetFace | 2.76 | 2.80 | 99.66 | 0.65 | 6.30 | | | | | | |

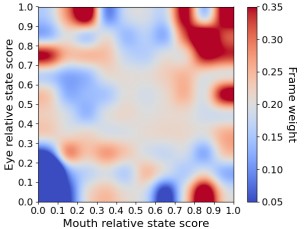

Figure 3: Heatmap of frame weights.

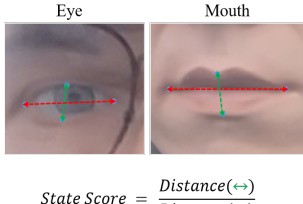

$$State\ Score\ = \frac{Distance(\leftrightarrow)}{Distance(\leftrightarrow)}$$

Figure 4: Diagram of eye and mouth state score calculation.

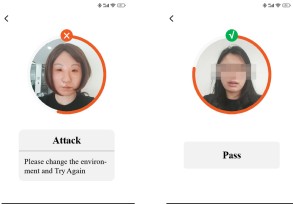

Figure 5: Screenshots of a 3D mask attack and a real face on mobile deployment.

the temporal information not being able to effectively guide spatial features. Hence, the AUC (99.55) is lower than when only spatial features are considered (99.64), and the HTER is 19.9% higher. When $\lambda$ rises to 8 or 10, although more temporal information brings about marginal improvement compared with $\lambda = 2$, processing more pairs of optical flows requires a larger computational budget and longer inference time. When there are no strict requirements for accuracy or inference time, variants of FASTEN with different $\lambda$ values are also practicable.

**Contribution of flow attention.** To further demonstrate the effectiveness of flow attention, we plot the relationship between facial relative state scores and frame weights as a heatmap in Figure 3. Given $n$ videos, we first get the eye/mouth state score $s_{ij}$ of the $j$-th frame in the $i$-th video by dividing the height of the eye/mouth by their corresponding width (as shown in Figure 4). The eye state score is averaged between both eyes. Next, we calculate the set $\mathbf{S}$ of relative eye/mouth state scores:

$$\mathbf{S} = \phi \left( \bigcup_{i=1}^{n} \left\{ \frac{e^{s_{ij}}}{\sum_{l=1}^{\lambda} e^{s_{il}}} \right\}_{j=1}^{\lambda} \right), \tag{3}$$

where $\phi(\cdot)$ is a truncation-normalization function that retains the values between the $5\% - 95\%$ distribution of all $n \times \lambda$ values and then normalizes them to $[0, 1]$. Larger eye/mouth relative state scores represent the relative trend of closing eyes and opening mouths. Similar to intuition, we observe that frames with larger action amplitudes tend to have higher frame weights. FASTEN pays the greatest attention to these frames with both mouth opening and eye closing (upper right corner). We also observe that the frame weight is high for frames in which subjects are about to open their mouths widely or close their eyes. Considering that blinking happens very quickly and the opening of the mouth is not always fully open, the maximum weight value is not at the maximum score in these cases (upper left corner and lower right corner). Overall, flow attention can effectively help FASTEN assign larger frame weights to frames with large movement.

## 4.4 Real-world deployment in mobile devices

To show the inference performance of our proposed FASTEN in resource-constrained platforms (*e.g.,* mobile devices), we package our model into an Android application and deploy it to the mobile end in the real world. Figure 5 shows two

Table 5: Mobile deployment performance.

| Device | Processor | GPU | Time | Storage | Memory |
|---|---|---|---|---|---|
| Samsung S8 | Snapdragon 835 | Adreno 540 | 224ms | 9.21MB | 13.89MB |
| Xiaomi 8 | Snapdragon 845 | Adreno 630 | 145ms | 9.21MB | 10.02MB |

screenshots of mobile deployment. The test experiments are conducted on a Samsung and a Xiaomi smartphone. Table 5 shows that FASTEN only consumes 9.21MB for storage, with a required run-time memory of less than 14MB. Furthermore, FASTEN achieves an average inference time of less than 224ms per image. We do not use batch processing when inferring. Considering the pre-processing time for video parsing, face detection with landmark, and face alignment (no post-processing), the overall response time is about 879 ms. Due to the low computational cost and model

parameters, quantization acceleration is unnecessary when deploying on mobile devices, so we can practically use it with almost no loss of accuracy. The overall system achieves a defense accuracy of 99.3% against 3D masks, indicating that our proposed framework can realize practical 3D mask PAD.

## 5   Discussion on research focus

Detecting 3D masks has become a rising challenge for most existing PAD methods owing to the rapid development and maturity of 3D printing [22]. Realistic 3D masks are hard to differentiate even for human eyes. To date, effective defenses against 3D masks remain a missing piece. Therefore, our method focuses on this type of attack. As a deep learning method by extracting features, we believe that our method can also be adapted to other attack types, *e.g.,* printing, replay, since their features are relatively simpler. Moreover, in actual face anti-spoofing security applications, it is usually an aggregation of multiple defense models. Other attacks can be mitigated separately using the corresponding tactics. Deploying our defense will enhance the overall security of the system.

## 6   Conclusion

This paper innovatively proposes a 3D mask detection framework, FASTEN, which addresses the generalizability deficiency of HFMs and DLMs and the high computational complexity of rPPGMs. Mainly including a facial optical flow network, FlowNetFace, a flow attention module, and a spatio-aggregation process, FASTEN fully utilizes spatial and temporal information to assign different frame weights to each frame. In addition, FASTEN only needs any five frames in sequential order from the RGB video as input, which can reduce computational overhead. FASTEN shows excellent superiority and generalizability compared with eight competitors in both intra-dataset and cross-dataset evaluations. Furthermore, we achieve real-world practical detection for 3D masks by releasing an application to mobile devices, which also illustrates the landing feasibility and high potential of our proposed method for practical usage.

**Limitation and future work.** The aforementioned contribution of flow attention has demonstrated that focusing on frames with more obvious movement details works well for detection. Actually, the distinguishability of different regions within each frame also varies. In future work, we will be dedicated to exploring inter- and intra-frame attention to further improve detection accuracy.

## 7   Broader Impact

Our proposed defense alleviates 3D mask spoofing attacks against face recognition systems (FRS). Such defense would be beneficial in minimizing the security concerns associated with FRS in the financial and public service sectors. However, we would also like to extend our considerations to privacy and fairness issues of the defense and its serving FRS. 1) Privacy. Our defense applies to the specific task of spoofing detection in which facial information is presented to the associated FRS as credentials. To ensure that no private information is memorized, our model is trained on public datasets and does not use any private photos for training purposes. 2) Fairness. In our work, we use HiFiMask as one of the main datasets. HiFiMask provides an equal number of subjects (25 each) for each ethnicity to facilitate fair artificial intelligence and mitigate bias [29]. Our training method also encourages the fairness of the trained defense. However, considering that defense is an independent component in the FRS, it cannot improve the fairness of the overall FRS. Moreover, there might be fairness issues for newborns or people with certain skin diseases, since they do not have enough representation in the training dataset. Another possibility would be that the dataset itself contains unbalanced data or faces with even one single skin tone. If this happens, incremental training after collecting obfuscated/synthetic data of the corresponding group of people can be used to mitigate the fairness issue. We also encourage future research in this area to incorporate fairness training techniques to mitigate possible issues.

## 8   Acknowledgement

This work was supported in part by facilities of Ping An Technology (Shenzhen) Co., Ltd., China and CSIRO's Data61, Australia. Minhui Xue and Derui Wang are the corresponding authors.

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
