# OpenReview forum: "Flow-Attention-based Spatio-Temporal Aggregation Network for 3D Mask Detection"
_NeurIPS.cc/2023/Conference — NeurIPS 2023 poster_

### Official Review · Reviewer_jUZR · 2023-06-12

**Soundness:** 3 good
**Presentation:** 3 good
**Contribution:** 2 fair
**Rating:** 6
**Confidence:** 4

**Summary:**

This paper presents a framework called FASTEN (Flow-Attention-based Spatio-Temporal Aggregation Network) for 3D mask presentation attack detection. In previous works, as a recent technology rPPG addresses some of the limitations but also sensitivity to noise and high computational overhead.

To overcome these challenges, FASTEN is designed to focus on fine-grained details in large movements, aiming to eliminate redundant spatio-temporal feature interference and capture splicing traces of 3D masks using fewer frames.

It comprises three key modules: a facial optical flow network for obtaining inter-frame flow information, flow attention for assigning different significance to each frame, and spatio-temporal aggregation for combining high-level spatial features and temporal transition features.

**Strengths:**

Through extensive experiments, FASTEN demonstrates a good performance compared to six competing methods in both intra-dataset and cross-dataset evaluations using multiple detection metrics.

The three datasets used for the evaluation are frequently used in the field and are publicly available.

The authors mention the deployment of FASTEN on real-world mobile devices for real-time 3D mask detection. This demonstrates the practicality and applicability of the proposed framework in real-world scenarios, validating its potential for deployment in various practical applications and settings.

**Weaknesses:**

The proposed method by the authors seems to have limited novelty as a substantial portion of its components is derived from existing work.

The experiments and discussions in real-world scenarios are very limited.

**Questions:**

The proposed method seems to focus on fine details in large movements. However, I am very curious about how the model will perform if the detected object does not have large movements.

In line 44, As far as I know, the majority of rPPG methods involve analyzing facial videos captured by a camera and do not necessitate the use of additional photodetectors.


**Limitations:**

It might be interesting if we could see more discussions about the different regions.

It would be useful to see more experiments and discussions in real-world scenarios. For example, a comparison of performance with previous methods in real-world scenarios.

---

> ### Author Rebuttal · Authors · 2023-08-10
>
> Reviewer jUZR:
>
> Q1: Limited novelty as a substantial portion of its components is derived from existing work.
>
> A1:  Simply using both spatial and temporal features is out of intuition and not new for detection tasks, but our contribution focuses more on the frame-wise attention to aggregate the feature information of multiple frames, rather than simple addition or concatenation of the two features. We provide comprehensive experimental results in the ablation study (Table 4) to show the outperformance of our defense (Spat. + Temp.(UnEq.)) over only combining two features together (Spat. + Temp.(Eq.)) which neglects the apparent movement changes among different frames. Assigning different frame weights in our method contributes more to the detection with significant improvement, especially when the input frames include large movements, e.g., eye-blinking, mouth-opening. According to our best knowledge, we are the first to consider flow attention and spatio-temporal aggregation for 3D mask detection. We will add more explanation in the revised manuscript.
>
> Q2: The experiments and discussions in real-world scenarios are very limited.
>
> A2: Our work proves that the accuracy is almost unaffected when the model is deployed in real-world scenarios. Considering the development cost and time constraints, we devote more effort to comparing and discussing experimental data.
>
> Q3: Large movements and different regions.
>
> A3: Our method performs better for input videos with larger motions, but since we also have a spatial branch, when the motion range is small, our method will rely more on the spatial branch, with the temporal branch providing assistance. Our experimental results show the average results of the whole dataset, which includes both large movements and trivial movements. The sound results can provide effective support for our method's effectiveness in inputs with subtle movements. We think it’s not scalable to conduct a comparison experiment to test the defense performance solely in small movements of the dataset since it is hard to quantitatively define the boundary between large and trivial movements.
>
> Q4: Error in line 44.
>
> A4: We admit this is a factual error. We will modify it in the revised version.

---

> > ### Comment · Reviewer_jUZR · 2023-08-13
> >
> > The authors' rebuttal has clarified my questions.

---

### Official Review · Reviewer_iaxD · 2023-07-03

**Soundness:** 3 good
**Presentation:** 3 good
**Contribution:** 3 good
**Rating:** 3
**Confidence:** 5

**Summary:**

In this paper, To enhance the accuracy of face presentation attack detection by effectively incorporating temporal information，the author proposes a novel 3D mask detection framework. The architecture integrates: 1) a facial optical flow network to obtain non-RGB inter-frame flow information; 2) flow attention to assign different significance to each frame; 3) spatio-temporal aggregation to aggregate high-level spatial features and temporal transition features. Through extensive experiments, FASTEN only requires five frames of input and outperforms six competitors for both intra-dataset and cross-dataset evaluations in terms of multiple detection metrics.

**Strengths:**

1.The author employs a simple model structure that enables deployment on end devices. It is hoped that the author can further compare the Flops and Params of this model with other models.
2.The author simplifies the structure of the optical flow calculation model specifically for facial features, thereby reducing computational burden.
3.The model only requires five images as input, reducing the requirement for the length of the input sequence compared to the rPPG method.

**Weaknesses:**

1.The structure proposed in this article is not sufficiently innovative, as it only uses simplified optical flow networks and MobileNetV3-Small. The fusion of the spatial features and optical flow features is achieved by employing the "calculate the frame weights" method.
2.Due to this paper has achieved results beyond previous works by using a simple structure, it is hoped that the author will open-source the code for validation purposes.
3.Although the model exhibits significant improvement in accuracy compared to rPPG-based models, which is attributed to the longer temporal information required for measuring rPPG signals, the improvement in accuracy compared to "Deep-Learning-based Methods" from two years ago is limited. It is requested that the author compare their approach with recent algorithms developed in the past two years.

**Questions:**

none

**Limitations:**

see weakness

---

> ### Author Rebuttal · Authors · 2023-08-10
>
> Reviewer iaxD:
>
> Q1: Novelty.
>
> A1: Simply using both spatial and temporal features is out of intuition and not new for detection tasks, but our contribution focuses more on the frame-wise attention to aggregate the feature information of multiple frames, rather than simple addition or concatenation of the two features. We provide comprehensive experimental results in the ablation study (Table 4) to show the outperformance of our defense (Spat. + Temp.(UnEq.)) over only combining two features together (Spat. + Temp.(Eq.)) which neglects the apparent movement changes among different frames. Assigning different frame weights in our method contributes more to the detection with significant improvement, especially when the input frames include large movements, e.g., eye-blinking, mouth-opening. As mentioned by Reviewer U1im, and also according to our best knowledge, we are the first to consider flow attention and spatio-temporal aggregation for 3D mask detection. We will add more explanation in the revised manuscript.
>
> Q2: Open source.
>
> A2: We feel sorry that we currently cannot open-source our code for public disclosure since it may involve confidentiality. We will open-source a demo code and model structure once the paper is accepted.
>
> Q3: It is requested that the author compare their approach with recent algorithms developed in the past two years.
>
> A3: We additionally compare our method with one ViT-based method, ViTranZFAS [1], and one deep-learning-based method, MD-FAS [2].
> Table 1 in the enclosure shows the intra-dataset and cross-dataset results on HiFiMask. ViTranZFAS performs worse than FASTEN. We attribute it to three main reasons: 1) Lack of Temporal Information: ViT primarily focuses on analyzing individual images without considering temporal information. Facial liveness detection often requires understanding the dynamic changes over time, such as blinking, movement, or facial expressions. ViT's static image analysis might not effectively capture these temporal cues. 2) Small Regions of Interest: Liveness cues often involve subtle facial regions like the edge of the facial features or muscle contractions. ViT's patch-based approach might not focus on these small regions of interest effectively, leading to challenges in capturing fine-grained liveness indicators. 3) Training Difficulty: ViT requires a pretrained model, which requires high computational costs.
>
> MD-FAS aims to solve the forgetting issue when learning a new domain data and improve the adaptability of the model. It cannot perform well with unseen target data, as shown in the cross-dataset evaluation. In addition, the baseline SRENet selected by this scheme is a region-based fine classification task, which has high requirements for the diversity of attack categories, as well as the resolution of images. When we use HiFiMask as the training set for a fair comparison, its accuracy performance remains poor.
>
> Reference:
>
> [1] George A, Marcel S. On the effectiveness of vision transformers for zero-shot face anti-spoofing[C]//2021 IEEE International Joint Conference on Biometrics (IJCB). IEEE, 2021: 1-8.
>
> [2] Guo X, Liu Y, Jain A, et al. Multi-domain Learning for Updating Face Anti-spoofing Models[C]//European Conference on Computer Vision. Cham: Springer Nature Switzerland, 2022: 230-249.

---

### Official Review · Reviewer_U1im · 2023-07-06

**Soundness:** 4 excellent
**Presentation:** 4 excellent
**Contribution:** 3 good
**Rating:** 7
**Confidence:** 4

**Summary:**

This work proposes a 3D mask detection system, successfully deployed on mobile devices. In addition, the number of frames required for making predictions is minimal, improving the latency of responses. Experiments,  performed with face mask datasets showed improved performance compared to existing techniques.

**Strengths:**

1. The introduced method is very interesting.
2. Combination of facial optical flow for inter-frame characteristics with flow attention and spatio-temporal aggregation (to the best of my knowledge) hasn’t been applied for 3D face mask detection yet.
3. The use of only 5 frames and potential of deploying such solution on mobile devices is very promising.
4. The method is well described with clear comparison to hand-crafted features, deep learning methods and remote vital estimation approaches.
5. Experimental analysis is clear and thorough with results supporting claims well.

**Weaknesses:**

1. The method has only been tested using MobileNet backbone. It’s understandable that experimental analysis has to have some limitations, but it would be interesting to explore other maybe even faster classifiers.
2. Minor comment: Font in tables is very small, please increase readability.

**Questions:**

1. How would motion affect performance of the proposed method? You stated that ‘flow attention can effectively help FASTEN assign larger frame weights to frames with large movement.’ but what type of the movement is acceptable?
2. Average processing time has been provided, but what’s the latency of responses? Do you use batch processing at the inference time?

**Limitations:**

Limitations and ideas for future work improvement were clearly explained and make sense.

---

> ### Author Rebuttal · Authors · 2023-08-10
>
> Reviewer U1im:
>
> Q1: More backbones.
>
> A1: Thanks for your advice. We use MobileNet due to its excellent accuracy-efficiency trade-off. Replacing it with other backbones may accelerate the speed but the performance improvement is limited. We will consider this advice in our future work.
>
> Q2: Fonts in tables.
>
> A2: We will check the fonts in all tables and figures and adjust them for better readability.
>
> Q3: Large movement.
>
> A3: One of our work’s insights is that the motion characteristics difference between the mask and that of the face, due to that the mask is not perfectly fit for the face. In practical applications, we consider movements such as nodding, shaking the head, and opening the mouth.
>
> Q4: Latency and batch processing.
>
> A4: The time mentioned in Table 5 refers to the time it takes for a single image to be processed using a mobile device. Considering the pre-processing time for video parsing and other necessary steps in actual usage, the overall response time is about 879 ms. We do not use batch processing at the inference time.

---

> > ### Comment · Reviewer_U1im · 2023-08-13
> > **Response to authors' rebuttal**
> >
> > Thank you for your responses. If the overall time of processing the entire pipeline is 879ms, then the method is not real time as stated in the paper. Please correct such statements. Please also include details of pre/post processing.

---

> > > ### Author Response · Authors · 2023-08-14
> > >
> > > Thanks for your reminder. We will modify the real-time issue in the revised manuscript. Pre-processing includes video frame decoding, face detection with landmark, and face alignment. There is no post processing.

---

### Official Review · Reviewer_bEHS · 2023-07-07

**Soundness:** 2 fair
**Presentation:** 3 good
**Contribution:** 2 fair
**Rating:** 3
**Confidence:** 5

**Summary:**

This paper presents an approach to 3D mask detection using a small number of face video frames. Experiments were conducted on several databases, and comparisons with other methods are presented.

**Strengths:**

Using small networks for 3D mask detection and a small number of frames;

Both spatial and temporal features are extracted;

Most parts are written clearly.

**Weaknesses:**

It seems that this work only focuses on 3D mask detection, without dealing with paper print, video replay, etc. This makes it quite narrow in terms of applications for face anti-spoofing.

It is not new to use both spatial and temporal features for face anti-spoofing. Many existing works have used spatial and temporal information.

More recent works use vision transformer based approaches for face anti-spoofing, which can effectively encode attention into the learning framework in a simple way, rather than using optical flow to indicate attention, as shown in the work. Further, there are no comparisons with the ViT based approaches in this paper.

It is confusing for the saying of "only 5 frames", since the five frames can be picked from a sequence with some fixed intervals, rather than consecutive frames in the original videos (25 or 30 frames per second), as indicated in some databases. Further, the paper argues to capture facial changes caused by pulse, how to acquire the blood changes within 5 frames? You can calculate: 5/25=0.2 seconds, which is less than a pulse period (suppose a regular person has 60 pulses per minute, then one pulse will take one second).

It mentions that the method can be developed in a mobile device, and shows the computation and memory used. But there is no accuracy or error rate reported. How accurate it can achieve in mobile device?

**Questions:**

See my questions and comments in Weakness part.

**Limitations:**

The argument of using 5 frames is doubtable.

---

> ### Author Rebuttal · Authors · 2023-08-10
>
> Reviewer bEHS:
>
> Q1: Only focusing on 3D mask detection, which is quite narrow in terms of applications for face anti-spoofing.
>
> A1: Owing to the rapid development and maturity of 3D printing, detecting 3D masks has become a rising challenging task in the field of face anti-spoofing, which threatens most existing PADs [1][2][3]. 3D masks made up of soft silicone are vivid enough, in terms of color, texture, and geometry structure, that even humans find it confusing to differentiate. This has gradually made 3D mask defense from a branch of face anti-spoofing to a relatively independent sub-topic. To date, effective defense against 3D masks remains a missing piece. Therefore, our method focuses on this type of attack (this is also where the motivation of our method comes from). As a deep learning method by extracting features, we believe that our method can also be used in other attack types e.g., printing, replay, since their features are relatively simpler. In addition, in terms of the actual application you mentioned, in actual face anti-spoofing security applications, it is usually the aggregation of multiple defense models. Other attacks can be separately mitigated by the corresponding tactics. Deploying our defense model will enhance the overall security of the system.
>
> Q2: It is not new to use both spatial and temporal features for face anti-spoofing.
>
> A2: Simply using both spatial and temporal features is not new for detection tasks, but our contribution focuses more on the frame-wise attention to aggregate the feature information of multiple frames, rather than simple addition or concatenation of the two features. We provide comprehensive experimental results in the ablation study (Table 4) to show the outperformance of our defense (Spat. + Temp.(UnEq.)) over simply combining two features together (Spat. + Temp.(Eq.)) without considering frame weights. According to our best knowledge, we are the first to consider flow attention and spatio-temporal aggregation for 3D mask detection. We will add more explanation in the revised manuscript.
>
> Q3: Recent works using ViT, can effectively encode attention into the learning framework in a simple way, rather than using optical flow to indicate attention.
>
> A3: The “attention” you mentioned here is different to the form of attention in our method. We used the optical flow to generate inter-frame attention (the contribution or weight of each frame to the detection) but not the attention on each feature map. However, we reckon this comment raised an interesting point, the attention on the spatial dimension also plays an important indication to the final detection. We therefore compare our defense with a ViT-based method, ViTranZFAS [4], which is slightly modified to adapt to 3D mask detection. Table 1 in the enclosure shows the intra-dataset and cross-dataset results on HiFiMask. FASTEN performs better than ViTranZFAS in almost all cases. We attribute it to three main reasons: 1)  Lack of Temporal Information: ViT primarily focuses on analyzing individual images without considering temporal information. Facial liveness detection often requires understanding the dynamic changes over time; ViT's static image analysis might not effectively capture these temporal cues. 2) Small Regions of Interest: Liveness cues often involve subtle facial regions like the edge of the facial features or muscle contractions; ViT's patch-based approach might not focus on these small regions of interest effectively, leading to challenges in capturing fine-grained liveness indicators. 3) Training Difficulty: ViT requires a pretrained model and high computational costs.
>
> Q4: “Only 5 frames”
>
> A4: Our method supports picking 5 frames at equal/unequal intervals. For datasets such as HiFiMask that have been pre-sampled, we directly use the preprocessed sampled data, since our method is not sensitive to how the 5 frames are selected, and just focuses on the movement between consecutive selected frames. We explained in the Statement for rPPG methods in the paper that they usually require a longer video clip (usually more than 10 seconds), and require continuous (high-frequency sampling, by Nyquist–Shannon Sampling Theorem) signal inputs. So “5-frame input” is for our method, not rPPG methods.
>
> Q5: How accurately can it be achieved in mobile devices?
>
> A5: With less computational cost and model parameters, quantization acceleration is unnecessary when deploying on mobile devices, so we can practically use it with almost no loss of accuracy. The overall system achieves a defense rate (accuracy) of 99.3% against 3D masks.
>
> Reference:
>
> [1] Liu S Q, Lan X, Yuen P C. Learning Temporal Similarity of Remote Photoplethysmography for Fast 3D Mask Face Presentation Attack Detection[J]. IEEE Transactions on Information Forensics and Security, 2022, 17: 3195-3210.
>
> [2] Yu Z, Qin Y, Li X, et al. Deep learning for face anti-spoofing: A survey[J]. IEEE transactions on pattern analysis and machine intelligence, 2022, 45(5): 5609-5631.
>
> [3] Erdogmus N, Marcel S. Spoofing face recognition with 3D masks[J]. IEEE transactions on information forensics and security, 2014, 9(7): 1084-1097.
>
> [4] George A, Marcel S. On the effectiveness of vision transformers for zero-shot face anti-spoofing[C]//2021 IEEE International Joint Conference on Biometrics (IJCB). IEEE, 2021: 1-8.

---

### Official Review · Reviewer_VV4D · 2023-07-07

**Soundness:** 3 good
**Presentation:** 2 fair
**Contribution:** 2 fair
**Rating:** 5
**Confidence:** 1

**Summary:**

The paper presents a 3D mask detection tool called FASTEN, aimed at making face recognition systems more secure. The proposed network focuses on fine-grained details in large movements, which helps eliminate redundant spatio-temporal feature interference.  This approach lowers computational overhead and it outperforms six competitive models in both intra-dataset and cross-dataset evaluations as Table 2/3.  Real-world application on mobile devices (Table 5) shows the efficiency of the method.

**Strengths:**

1. FASTEN offers a solution by focusing on small details during big movements, helping to detect traces of 3D masks in fewer frames.
2. The reduction in frames needed from 25 to just 5 means this tool is computationally more efficient and suitable for real-time applications.
3. The fact that FASTEN can be used in real-time on mobile devices shows that it's practical for everyday use.

**Weaknesses:**

I am not directly working on the research topic. The contribution of this paper is more technical that combines the FlowNet and mobilenetv3 (as Fig. 2) to improve the efficiency of 3D mask detection across 5 frames. I think the authors should state the explicit insights of this paper, compared to the simple baseline of FlowNet+mobileNetv3.

**Questions:**

See the weakness.

**Limitations:**

N/A.

---

> ### Author Rebuttal · Authors · 2023-08-10
>
> Reviewer VV4D:
>
> Q1: I think the authors should state the explicit insights of this paper, compared to the simple baseline of FlowNet+mobileNetv3.
>
> A1: It is intuitive to consider both spatial and temporal features when detecting 3D masks. However, as shown in the ablation study (Table 4), simply combining these two (Spat. + Temp.(Eq.)) will neglect the apparent movement changes among different frames, thus leading to average performance. Instead, we introduce frame weights obtained by temporal features to assign different significance to the spatial features in each frame. This contributes more to the detection with significant improvement, especially when the input frames include large movements, e.g., eye-blinking, mouth-opening. We will clarify more explanation in the revised manuscript.

---

> > ### Comment · Reviewer_VV4D · 2023-08-18
> > **Thank you for your responses.**
> >
> > Thank you for your responses. As I am not an expert in this topic, I do not have more comments.

---

### Author Rebuttal · Authors · 2023-08-10

Thanks for your feedback! We summarize our responses by questions in this rebuttal. Please see more details in the responses to each reviewer.

* Ethics [Ethics Reviewer HeTH, Nmac]

We acknowledge there might be social impacts associated with our work. We will accordingly add a section to the paper to discuss the impact.

Our defense alleviates 3D mask spoofing attacks against face recognition systems (FRS). Such defense would be beneficial for minimizing security concerns associated with FRS in financial and public service sectors. However, we would also like to extend our considerations to privacy and fairness issues of the defense and its serving FRS. 1) Privacy. Our defense applies to the specific task of spoofing detection in which the facial information is presented to the associated FRS as credentials. To ensure no private information is memorized, our model is trained on public datasets and does not use any private photos for the training purpose. 2) Fairness. In our work, we use HiFiMask as one of the main datasets. HiFiMask provides an equal number of subjects (25 each) for each ethnicity to facilitate fair artificial intelligence and mitigate biases [1]. Our training method also encourages the fairness of the trained defense. However, considering the defense is an independent component in the FRS, it cannot improve the fairness of the overall FRS. Moreover, there might be fairness issues for newborns or people with certain skin diseases since they do not have enough representation in the training dataset. Another possibility would be the dataset itself contains unbalanced data or faces with even only one skin tone. If this happens, incremental training after collecting obfuscated/synthetic data of the corresponding group of people can be used to mitigate the fairness issue. We also encourage future research in this area to incorporate fairness training techniques to mitigate possible issues.

* Novelty [Reviewer VV4D, bEHS, iaxD, jUZR]

We emphasize that our contribution focuses more on the frame-wise attention to aggregate the feature information of multiple frames, rather than simple addition or concatenation of the two features. We provide comprehensive experimental results in the ablation study (Table 4) to show the outperformance of our defense (Spat. + Temp.(UnEq.)) over only combining two features together (Spat. + Temp.(Eq.)) which neglects the apparent movement changes among different frames. We are the first to consider flow attention and spatiotemporal aggregation for 3D mask detection.

* Research focus [Reviewer bEHS]

Detecting 3D masks has become a rising challenge for most existing PADs owing to the rapid development and maturity of 3D printing [2]. Realistic 3D masks are hard to differentiate even for human eyes. To date, effective defense against 3D masks remains a missing piece. Therefore, our method focuses on this type of attack (this is also where the motivation of our method comes from). As a deep learning method by extracting features, we believe that our method can also be adapted to other attack types e.g., printing, replay, since their features are relatively simpler. Moreover, in actual face anti-spoofing security applications, it is usually an aggregation of multiple defense models. Other attacks can be separately mitigated by the corresponding tactics. Deploying our defense will enhance the overall security of the system.

* More competitors [Reviewer bEHS, iaxD]

We additionally compare our method with ViTranZFAS[3], and one recent deep-learning-based method, MD-FAS[4]. Table 1 in the enclosure shows the intra-dataset and cross-dataset results on HiFiMask. Please refer to the responses to each reviewer for more analyses.

* Frames [Reviewer bEHS]

Our method supports picking five frames at equal/unequal intervals. For datasets such as HiFiMask that have been pre-sampled, we directly use the preprocessed sampled data, since our method is not sensitive to how the frames are selected, and just focuses on the movement between consecutive selected frames. We explained in the Statement for rPPG methods in the paper that they usually require a longer video clip (usually more than 10 seconds), and require continuous (high-frequency sampling, by Nyquist–Shannon Sampling Theorem) signal inputs. So “5-frame input” is for our method, not rPPG methods.

* Real-world scenarios [Reviewer bEHS, U1im, jUZR]

bEHS: With less computational cost and model parameters, quantization acceleration is unnecessary when deploying on mobile devices, so we can practically use it with almost no loss of accuracy. The overall system achieves a defense rate (accuracy) of 99.3% against 3D masks.

U1im: The time mentioned in Table 5 refers to the time it takes for a single image to be processed using a mobile device. Considering the pre-processing time for video parsing and other necessary steps in actual usage, the overall response time is about 879 ms. We do not use batch processing at the inference time.

jUZR: Our work proves that the accuracy is almost unaffected when the model is deployed in real-world scenarios. Considering the development cost and time constraints, we devote more effort to comparing and discussing experimental data.
* Open source [Reviewer iaxD]

We feel sorry that we currently cannot open-source our code for public disclosure since it may involve confidentiality. We will open-source a demo code and model structure once the paper is accepted.

Reference:

[1] Liu A, et al. Contrastive context-aware learning for 3d high-fidelity mask face presentation attack detection. TIFS, 2022.

[2] Yu Z, et al. Deep learning for face anti-spoofing: A survey. TPAMI, 2022.

[3] George A, et al. On the effectiveness of vision transformers for zero-shot face anti-spoofing. IJCB, 2021.

[4] Guo X, et al. Multi-domain Learning for Updating Face Anti-spoofing Models. ECCV, 2022.

---

### Decision · Program_Chairs · 2023-09-21

**Decision:**

Accept (poster)

**Comment:**

This paper presents an approach to 3D mask detection. Experiments on several databases demonstrated the effectiveness of the proposed approach. The author rebuttal provided feedback to the review comments. Three positive reviewer were satisfied with the rebuttal. The other two reviewers were negative and did not provide further comments after the rebuttal. The AC checked the comments from Reviewer iaxD and the feedback. The critical one is the third one, comparison to the recent algorithms is needed. The authors provided additional results, which are convincing. Regarding the comments from Reviewer bEHS, the AC checked the feedback and most comments were well addressed. One issue is about the accuracy for the mobile device. As the rebuttal said in the discussion with the reviewer, the authors will modify the description about the real-time issue. Please do make modification.